# Effect of Corrosion in the Transverse Reinforcements in Concrete Beams: Sustainable Method to Generate and Measure Deterioration

P. Castro-Borges [1], C. A. Juárez-Alvarado [2,*], R. I. Soto-Ibarra [2], J. A. Briceño-Mena [1], G. Fajardo-San Miguel [2] and P. Valdez-Tamez [2]

1 Centro de Investigación y de Estudios Avanzados del Instituto Politécnico Nacional (Center for Research and Advanced Studies of the National Polytechnic Institute, Merida Unit), Departamento de Física Aplicada (Department of Applied Physics), Unidad Mérida (CINVESTAV-IPN, Unidad Mérida) Km. 6 Antigua carretera a Progreso, Mérida C.P. 97310, Yucatán, Mexico; pcastro@cinvestav.mx (P.C.-B.); jorge.briceno@cinvestav.mx (J.A.B.-M.)

2 Facultad de Ingeniería Civil (Civil Engineering Faculty), Universidad Autónoma de Nuevo León (Autonomous University of Nuevo Leon), Av. Universidad s/n, Cd. Universitaria, San Nicolás de los Garza C.P. 66455, Nuevo León, Mexico; rsotoib@gmail.com (R.I.S.-I.); gerardo.fajardosn@uanl.edu.mx (G.F.-S.M.); pedro.valdeztz@uanl.edu.mx (P.V.-T.)

* Correspondence: cesar.juarezal@uanl.edu.mx; Tel.: +52-1-818-329-4000 (ext. 7220)

**Abstract:** A consistent method to generate and measure deterioration by corrosion in transverse reinforcements for concrete beams is presented and discussed in this work. This approach could be applied in other circumstances, such as bending, compression or combinations of stresses, with comparable results and therefore can be used to ensure sustainability. In marine environments, macro-cells are produced primarily from a transverse reinforcement, which works as an anode and therefore becomes a critical part of the structural analysis. To evaluate the adaptation efficiency of our proposed method, the corrosion potential, mass losses and cross-section reductions of the steel were measured. The shear stress behavior of the beams was investigated, including beam responses to load deformations, failure modes and cracking. The method ensured that all the beams exhibited a shear failure from diagonal stress with almost 50% less deflection when mechanically tested. The critical cross-sectional area, calculated according to the experimental diameter with the greatest cross-sectional loss due to the corrosion of the deteriorated stirrup, proved to be a reliable value for predicting the ultimate shear strength of concrete beams deteriorated by severe corrosion. A reduction of up to 30% in the shear strength of deteriorated versus non-deteriorated beams was found. Additional results showed that there is a correlation between the experimental and theoretical results and that the method is reproducible.

**Keywords:** corrosion; deterioration; stirrup; concrete; beams

## 1. Introduction

One of the primary problems associated with reinforced concrete elements exposed to marine environments is the attack of the reinforcing steel by chloride ions. The chloride ions enter the concrete either by diffusion, by absorption, by convection or by a combination of these processes [1]. Most often—and whatever the mode of entry—the stirrup is the reinforcement that is initially affected because of its proximity to the surface; however, the stirrup makes a significant contribution to the shear strength of a reinforced concrete beam. Therefore, the corrosion of the transverse reinforcement in concrete is an important topic for many researchers because of the mechanical impacts generated by the reduction

in the cross-section of the stirrups resulting from deterioration by corrosion. In the literature [2–5], several studies have focused on evaluating the mechanical properties and durability of this type of structural element. Some of these studies have assessed the effects of deterioration caused by corrosion in reinforced concrete beams over decades [6–8]. In most previous studies, the beams were exposed to aggressive environments for more than 20 years, which promoted the generalized corrosion of the reinforcements. These studies showed that modifying the rigidity and ultimate deformations has a significant impact on the mechanical behavior, decreasing the ductility of the beams [9]. However, because the entire reinforcement is affected, it is difficult to extract discriminated information from the existing mechanical phenomena. In this regard, studies have been performed to determine the effect of deterioration by corrosion in stirrups; specifically, the behavior of shear stress in concrete beams [10–12]. In these studies, accelerated corrosion was produced by inducing a current in the stirrups. Additionally, the specimens were submerged in salt solutions with different concentrations, resulting in decreases in the mechanical properties of the beams. Nevertheless, despite the similarities in the techniques and methodologies used, there were remarkable differences, primarily in terms of the methodology used to create corrosion to cause a shear failure. In some cases, cracks appeared in the bars longitudinally, showing potential corrosion damage. Similarly, the specimens exhibited different failure modes when testing the shear stresses of the beams. Previous studies showed results that, although extremely valuable, are difficult to compare because they were methodologically different, even though the effects of corrosion on shear reinforcements were studied. In this regard, searching for a comparison mechanism among studies, a previous study [13] attempted to isolate the shear strength using a diagonal stress on the beams to obtain deterioration results that were more comparable with those of other studies. The approach was different from previous studies; the beams were exposed to humidification and drying cycles using a 3.5% NaCl solution until the steel was de-passivated. Then, a current of 100 µA/cm$^2$ was applied for a period from 80 to 120 days to reach moderate and severe corrosion levels in the stirrups. The results showed a significant differentiation among the states of deterioration associated with a preponderant mechanical effect; i.e., shear stress. With this method, the influence on ductility (increase of fragility), loss of cross-section and remaining strength of the beams were studied to isolate the effect on the shear strength of different deterioration states. Although the literature can provide data representing the values of different parameters with sufficient credibility, the goal of our work is to obtain a method that is adaptable to common circumstances in all investigations and with different sources, which would ensure sustainability. For this reason—and with the intention of obtaining a simple but effective method for comparing the results of deterioration by corrosion in concrete beams—a differentiated method for inducing the effect of deterioration by the corrosion of stirrups, with respect to the shear strength in reinforced concrete beams, both theoretically and experimentally, is presented and discussed in this work.

## 2. Materials and Methods

The theoretical and experimental method are described separately below.

### 2.1. Theoretical Method

Theoretical Electrochemical Method for the Induction of Deterioration in Transverse Reinforcements

Some studies [14–17] have discussed the influence of a cross-sectional reduction on the decrease of tensile strength in the stirrups, which affects the ductility of the beams; these works also reported that a reduction by 20% of the cross-section of the stirrups represents severe deterioration, under which both the steel and the concrete are compromised.

In this study, a reduction of the transverse reinforcement diameter by 10% is taken as representative of the equivalent of a 20% reduction in the cross-section of the stirrup. Therefore, a maximum loss of stirrup diameter of 10% is considered to represent severe deterioration with respect to the impacts

of the stirrups on structural safety. The diameter loss is theoretically estimated as a measure of the remaining stirrup section with the following equation, as proposed by Andrade [18]:

$$\varnothing_t = \varnothing_i - 0.023 i_{corr} * t \qquad (1)$$

where

$\varnothing_t$ = change of diameter in time (mm);
$\varnothing_I$ = initial diameter of the bar (mm);
$i_{corr}$ = corrosion rate ($\mu$A/cm$^2$);
t = time after starting the propagation period (years);
0.023 = conversion factor used to convert $\mu$A/cm$^2$ to years.

The data shown in Figure 1 illustrate the values from Equation (1), using a proposed value of up to 200 $\mu$A/cm$^2$, to realistically reproduce severe corrosion damage and strain [19]. Corrosion products gradually dissipate with this current density level [20]. Higher values could produce oxide products that differ from real corroded structures, causing confusion in the interpretation of the results. In this way, the corrosion damage is theoretically estimated to predict the reduction in the section. For verification, a reduction of 10% of the diameter was established after 65 days of exposure.

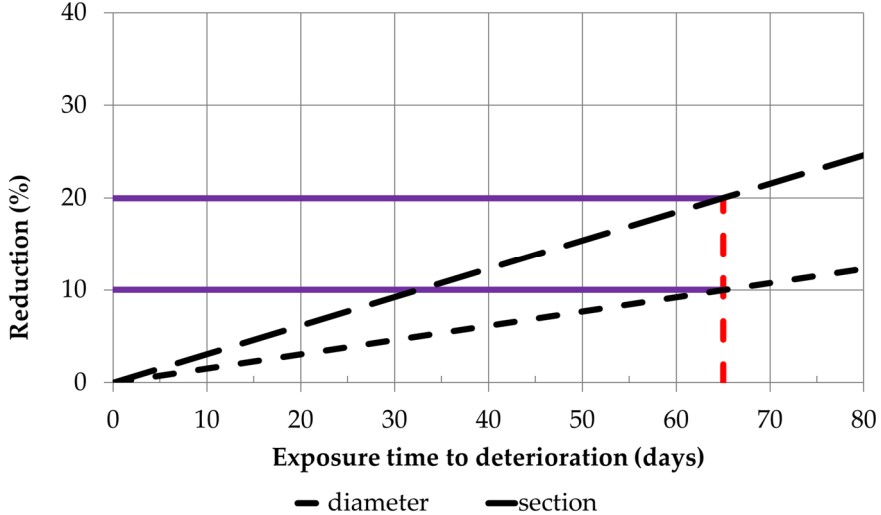

**Figure 1.** Loss of diameter and cross-section per time. $\varnothing_i$ = 8 mm, $i_{corr}$ = 200 $\mu$A/cm$^2$.

## 2.2. Experimental Method

The experimental and theoretical methods are linked; however, for the purposes of comparison and adaptation to other studies, the former could be modified to consider other circumstances.

### 2.2.1. Concrete Manufacturing

Table 1 is a summary of the mix used in the study. The concrete was manufactured under laboratory conditions with a water/cement ratio (w/c) of 0.55. Cement type CPC 30R was used, with aggregates of limestone. The maximum size of the coarse aggregate was 19 mm, and the fine aggregate passed through mesh No. 4 (mesh opening 4.76 mm). The resulting concrete reached the compressive strength rate in 28 days.

**Table 1.** Mix design.

| Cement | Water | Aggregate | | f'c | Slump |
|---|---|---|---|---|---|
| | | **Coarse** | **Fine** | | |
| 393 kg | 216 kg | 960 kg | 768 kg | 26 MPa | 150 mm |

### 2.2.2. Manufacturing the Beams

To evaluate the effect of the deterioration procedure of shear stress behavior, six reinforced concrete beams—i.e., two control beams with no deterioration (VSD 1 and 2) and four test beams with induced deterioration (VCD 1–4)—were manufactured; the dimensions of the reinforced concrete beams were set at 2000 mm × 200 mm × 350 mm. By design, a shear span (a) of 600 mm and an effective depth (d) of 296 mm were determined, thus obtaining a shear span to effective depth ratio a/d of 2.02—a value that represents a potential shear failure from diagonal stress. The structural arrangement is presented in Figure 2; the bending reinforcement was placed in two beds with a separation of 20 mm. The first layer comprised three bars, and the second comprised two bars, with diameters of 16 mm (Fy = 412 MPa) and number 2.5 bars with diameters of 8 mm (Fy = 460 MPa) for the stirrups. The longitudinal bars were coated with epoxy paint to avoid any damage by corrosion. The formation of galvanic stacks in the contact zones between the bending steel and the stirrups was prevented by the addition of electrical insulation tape. The traditional tie-up was replaced by nylon strips. The arrangement of the assembly is shown in Figure 3. The verification of this behavior is presented in Sections 4 and 5.

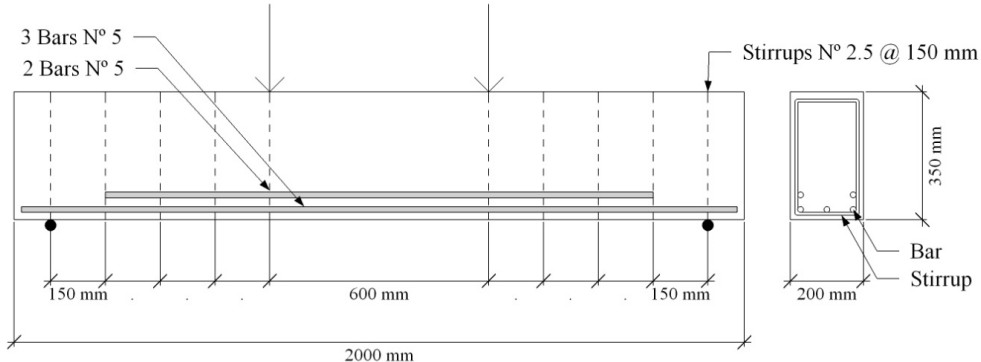

**Figure 2.** Structural arrangement to promote shear failure.

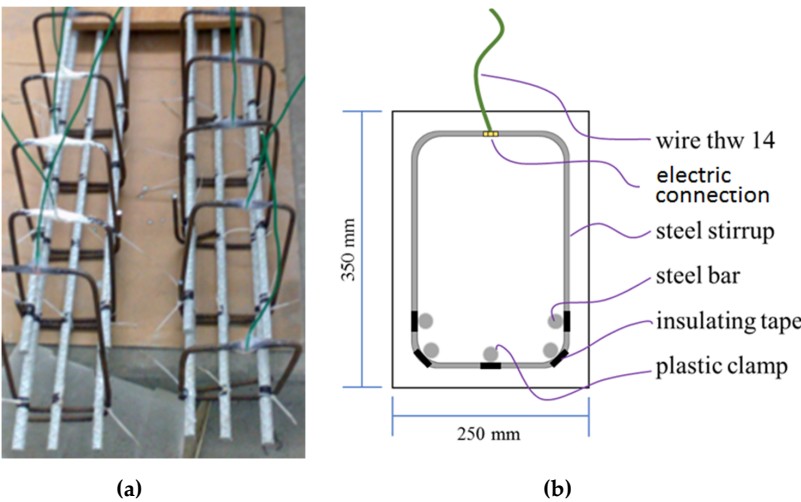

**Figure 3.** Assembly preparation for deterioration induction. (**a**) The reinforcement of the beam and its connections before being cast; (**b**) a diagram of the beam with details of connections for corrosion induction.

### 2.3. Method for Accelerated Deterioration Induction

Most significant results about structural behavior can be obtained by artificially corroded specimens [21]. Therefore, because of a wide and scattered spectrum of results for comparison purposes, a sustainable and reliable method to generate damage by severe corrosion in the reinforcement is needed. This is consistent and corresponds with the considerations proposed in the structural behavior described above for transversal rebars. This method must also comply with the conditions outlined in Figure 2. The method comprises the application of current, a humidification cycle and a drying cycle, as shown in Figure 4. The humidification of the beams is accomplished through polyurethane sponges, with dimensions of 600 mm × 300 mm × 20 mm, placed in the shear area. These sponges were moistened daily using a 3.5% NaCl solution to ensure that the applied current remained constant. After a week of humidification, a galvanic current was applied for 65 days to reach a severe deterioration level. The connection of each stirrup was made with a circuit formed by a rheostat of 0–50 kohm and a resistance of 1 ohm, with which it was possible to regulate the current applied to each stirrup. The circuit was connected to a power source of 0–30 volts and 0–5 amperes. A level of 200 $\mu$A/cm$^2$ was applied over the total area of each stirrup. The corrosion potential ($E_{corr}$) was monitored with an Ag/AgCl reference electrode, presented for analysis as $E_{corr}$ vs. Cu/CuSO$_4$ (CSE) in the corresponding charts. Measurements were taken weekly. In the following figures, the corrosion potentials are presented with respect to the days of effective exposure to deterioration (application of current, humidification cycle and drying cycle). The measurements of the potentials followed the standard procedures (ASTM C876 [22]).

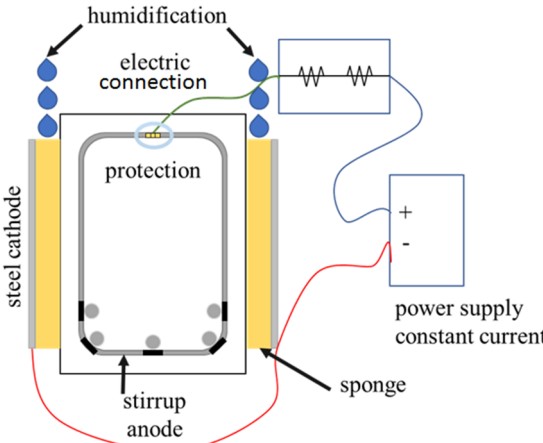

**Figure 4.** System for application of current, humidification and drying.

### 2.4. Mechanical Testing of the Beams

The simply supported beams were tested by applying concentrated loads to 600 mm sections of the supports, as shown in Figure 5. The test was carried out in a Tinius Olsen universal machine with a capacity of 200 tons. The total load was measured by a load cell with a capacity of 50 tons. The deflection at the midspan was measured with a linear variable displacement transducer (LVDT) with a calibration coefficient of 0.0005 cm. The shear strengths of the control beams without deterioration were determined at 28 days.

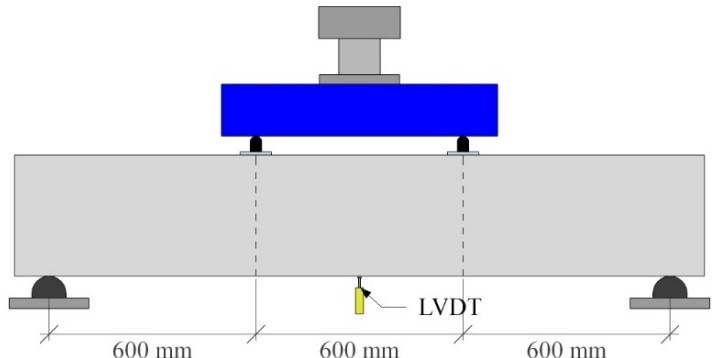

**Figure 5.** Testing scheme. LVDT: linear variable displacement transducer.

## 3. Results

### 3.1. Visual Inspection

In Figure 6, the visual appearance of the beam VCD-1 side A, after the induction of deterioration, is shown. The data are representative of all the beams. In Figure 7, the schematic appearance of the cracks generated in the stirrups during the process for the same beam are shown. The signs of corrosion—e.g., oxide stains and cracks (location, direction, and dimensions)—were recorded.

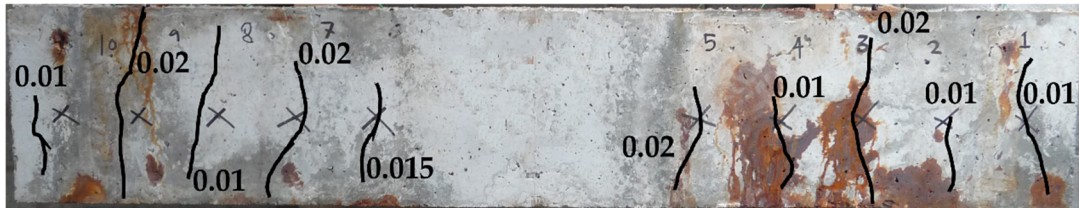

**Figure 6.** Visual appearance of beam VCD-1 side A after 65 days of applied deterioration; the crack widths are noted in cm.

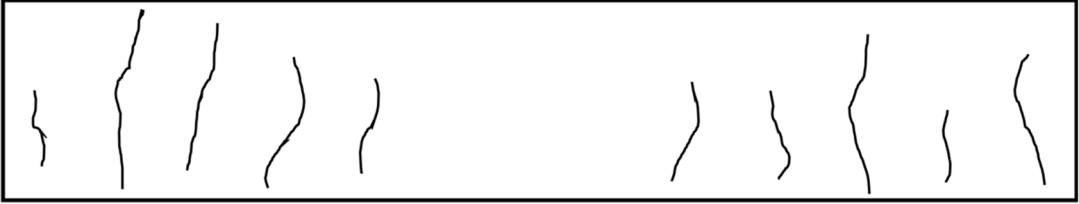

**Figure 7.** Cracking diagram for beam VCD-1 side A.

### 3.2. Measurement of the Corrosion Potentials

As shown in Figure 8, and for all the beams, the potential intervals indicated a negligible corrosion probability before the initiation of deterioration. Between 21 and 28 days after induction, potential levels vs. CSE were more negative than −350 mV; therefore, the beams were in a highly corrosive state. At the beginning of the process, the corrosion potentials of the stirrups showed some variability. This phenomenon is representative of the actual behavior of structures exposed to a marine environment, where there are variables that cannot be controlled, such as temperature and relative humidity. The potential levels measured at 50 and 65 days of exposure to induced deterioration reached values between −400 mV and −600 mV, respectively, indicating a high probability of advanced corrosion or severe deterioration, according to ASTM C876.

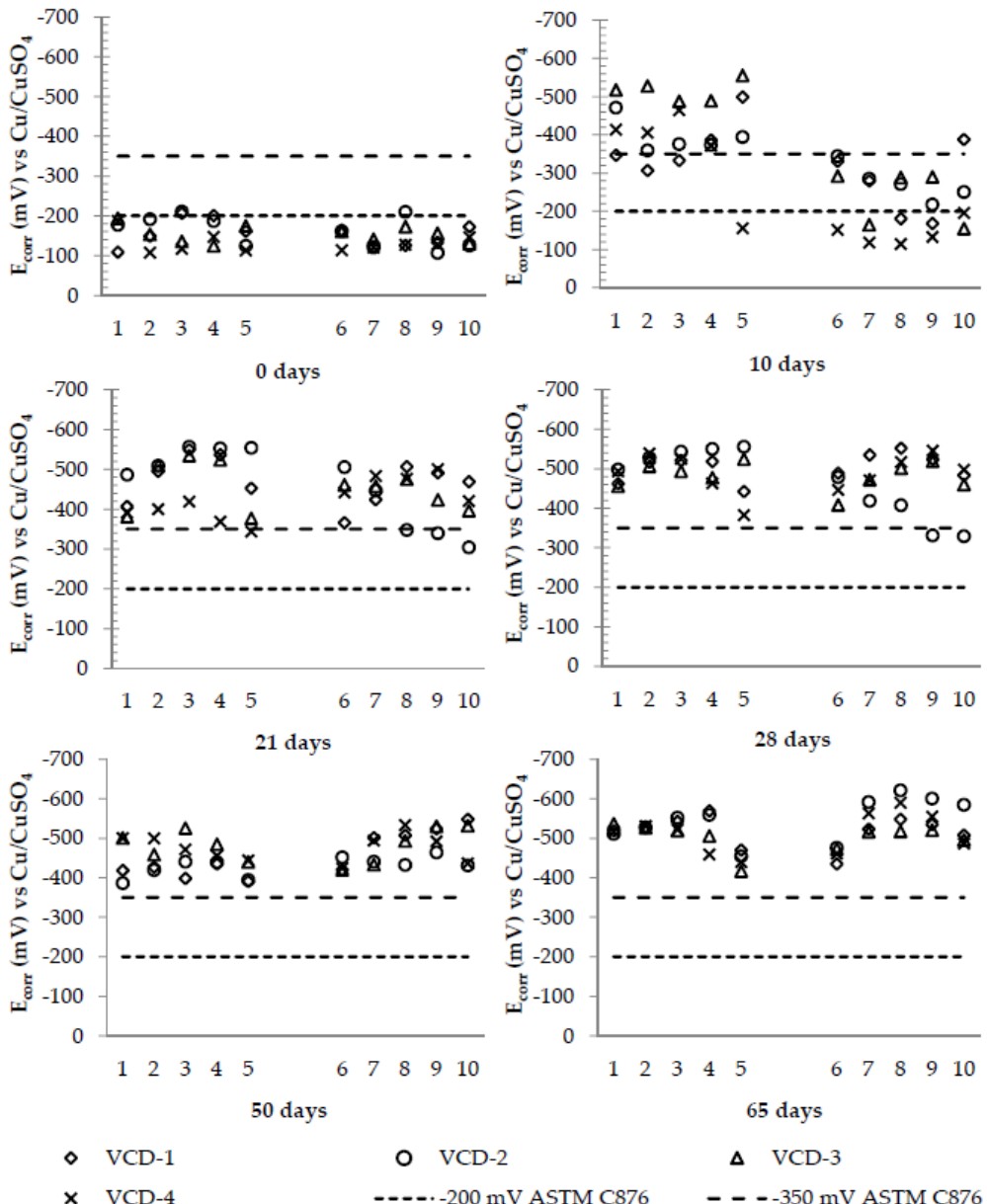

**Figure 8.** Electrochemical monitoring by measuring the corrosion potentials in the stirrups.

*3.3. Determination of Cross-Section and Mass Losses in the Transverse Reinforcements*

The previously tested beams were demolished to extract the corroded stirrups, as shown in Figure 9. Subsequently, the actual loss of the mass and cross-section of the transverse reinforcement was obtained [23]. Under the induced deterioration, the mass loss was as high as 11% on average. Some authors have reported that when there is a mass loss of 5–10% [24–26], the remaining capacities of the reinforced concrete beams and the steel/concrete adhesion decrease considerably. In Table 2, the relation between the theoretical values obtained by Equation (2) and the experimental values are shown.

$$W = \frac{ItM}{nF} \tag{2}$$

where

W = metal weight, g, which is corroded in an aqueous solution in the time t, in s;

I = current flux, in Amp;

M = atomic mass of metal, in g/mol;

n = number of consumed or produced electrons during the process;

F = Faraday constant, 96,500 C/mol.

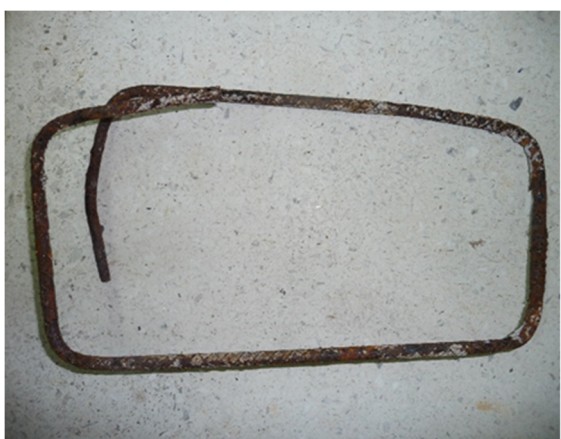

**Figure 9.** Removed corroded stirrup (position 8) with 65 days of exposure to induced deterioration.

**Table 2.** Relation between $\bar{x}$ (mass loss per stirrup) and the theoretical estimate.

| Beam | Experimental Mass Loss $\bar{x}$ | | Theoretical Mass Loss by Equation (2) | | Relation $\bar{x}$/ Equation (2) |
|---|---|---|---|---|---|
| | Stirrup | Percentage | Stirrup | Percentage | |
| VCD-1 | 43.0 g | 11.4% | 50 g | 12.50% | 0.86 |
| VCD-2 | 30.0 g | 8.0% | 50 g | 12.50% | 0.60 |
| VCD-3 | 40.0 g | 10.0% | 50 g | 12.50% | 0.80 |
| VCD-4 | 37.0 g | 10.0% | 50 g | 12.50% | 0.74 |

### 3.4. Determination of Chloride Content

In Table 3, the average concentration of chloride by weight of cement for severe deterioration is shown. The chloride concentration exceeded the maximum limit of total chlorides specified by ACI 318 [27] for structures exposed to a chloride environment [28]. Similarly, previous studies have shown that the threshold for total chlorides by weight of cement are from 1.24–2.15% [29,30]. The amount of total chlorides is a parameter that confirms the thermodynamic state of the transverse reinforcement steel, as evidenced by the $E_{corr}$ values in the beams.

**Table 3.** Chloride content.

| Beam | % Chloride by Weight of Cement |
|---|---|
| VCD-1 | 2.15 |
| VCD-2 | 3.00 |
| VCD-3 | 2.30 |
| VCD-4 | 2.75 |

### 3.5. Mechanical Resistance to Shear

The experimental results of shear structural behavior are presented in Figure 10 and Table 4, where the average deflection for the VSD beams is 11.4 mm. A variation in stiffness can be observed in VSD beams, although the beams are twins; there are factors that can cause such behavior, such as variations in the mixing, vibrating and curing of concrete, as well as different cracking kinetics

between the beams. However, the shear strength was very similar. In the VCD beams, the deflection at the midspan decreased, on average, by 8.0 mm; i.e., a 29.8% decrease in deflection. In general terms, the deterioration by corrosion in the shear reinforcements affected the beams' ductility, as expected, presenting a fragile and sudden failure; this was more evident in the VCD-1 and VCD-2 beams, which had a greater deterioration due to corrosion than the VCD-3 and VCD-4 beams, which caused a reduction in their cracking load. Similarly, the VCD beams decreased their average shear strength by 26% compared to the VSD beams. In Figure 11, the data show a cracking pattern at the end of the shear test. To avoid confusion, the corrosion cracks in Figure 11 were omitted.

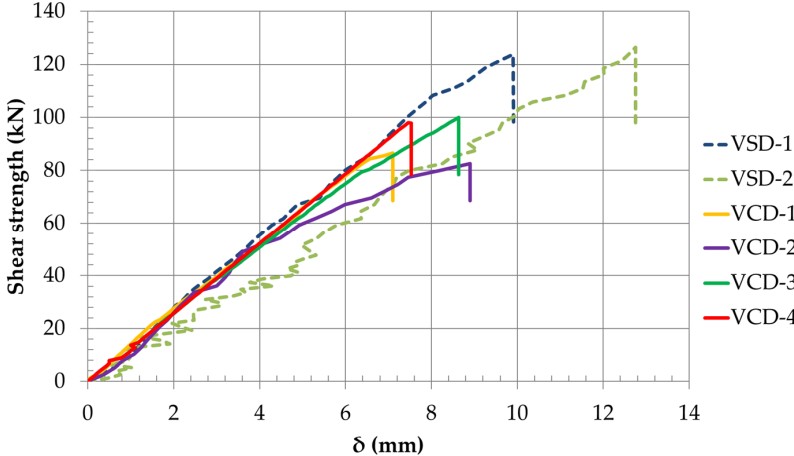

**Figure 10.** Shear strength–displacement curves.

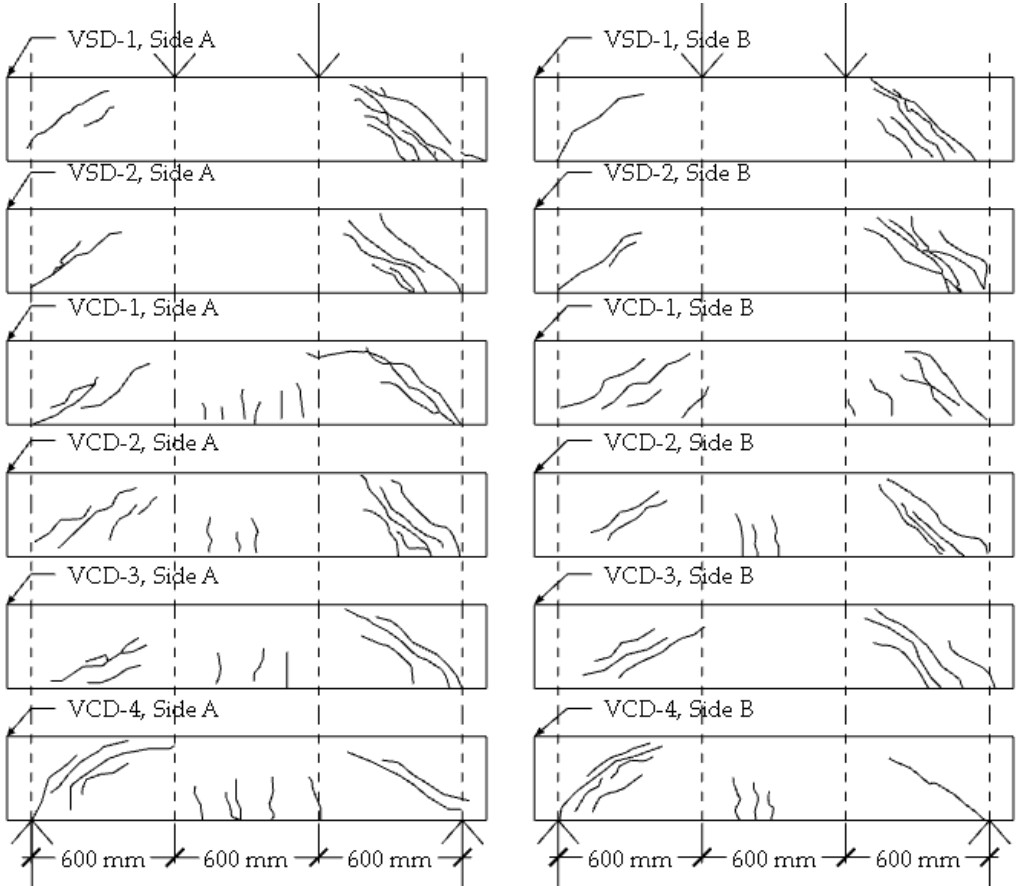

**Figure 11.** Cracking patterns obtained during the shear tests.

**Table 4.** Shear strength (Vu) and deflection at the midspan.

| VSD-1 | VSD-2 | VCD-1 | VCD-2 | VCD-3 | VCD-4 |
|-------|-------|-------|-------|-------|-------|
| 124 kN | 126 kN | 88 kN | 84 kN | 100 kN | 98 kN |
| 9.9 mm | 12.8 mm | 6.9 mm | 8.9 mm | 8.6 mm | 7.5 mm |

## 4. Discussion

### 4.1. Reliability of the Method for the Induction of Cracking in Concrete

A reliable method yields the expected results reproducibly. As the reproducibility increases, the reliability increases. The design of experiments for the types and sizes of specimens tested in this study is susceptible to variations in the manufacturing process, the instrumentation, exposure, the method of obtaining results, etc. More importantly, the nature of the induction of deterioration affects the design. In this study, the proposed adjustments to an induction of deterioration procedure, such as isolating the corrosion to a shear zone and making it consistent with what is expected from the structural calculation, show excellent reproducibility and can thus be used in future studies in different circumstances. Figure 12 is a schematic of the reproducibility of specific corrosion cracking patterns of the stirrup on all the beams, where cracks emerge on the concrete surface linearly to the reinforcement [19,31]. These results also show that there are neither micro-stresses in the concrete matrix that could have affected the contribution of the concrete central section nor concrete detachments in any area. This reproducibility was also manifested in the crack widths on the shear reinforcements whose values were in the range 0.1–0.2 mm, which is extremely narrow for experiments of this size. None of the cases showed coincident cracking to the bending of the reinforcing bars, as has been reported previously. A measure of the method's reliability to induce cracking is the number of stirrups that failed; in these tests, 36 of 40 stirrups—equivalent to 90% of the total stirrups—failed.

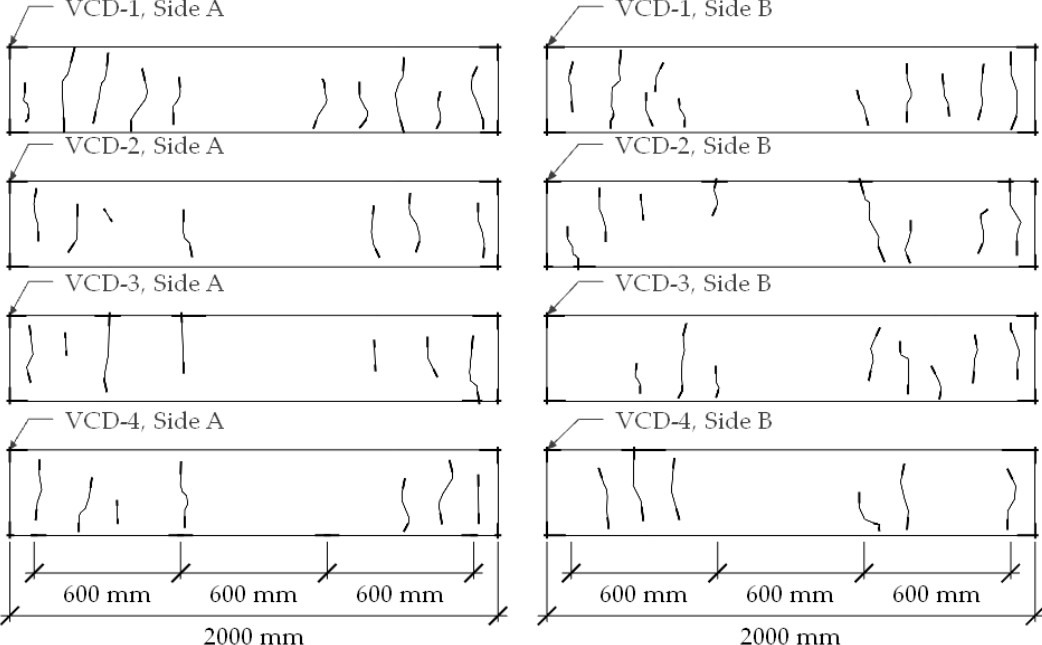

**Figure 12.** Cracking patterns of severe corrosion at the locations of the shear reinforcements.

### 4.2. Reliability of the Method to Generate Electrochemical Damage

The reproducibility was also verified through electrochemical monitoring. The evolution of the potentials with respect to time, shown in Figure 8, illustrates specific behavior. A homogenization of values at 65 days of exposure illustrates constant aggressiveness. Figure 13 is a representative

configuration of Figure 8. Figure 13 displays the electrochemical value at the location of the transverse reinforcement that identifies its tendency. Note also that the values of the corrosion potential, despite being a thermodynamic parameter, move in a narrow band below −350 mV vs. CSE. The exceptions were stirrups 5 and 6, which coincided with the ends of the sponges, demonstrating that the attack method must have extreme continuity.

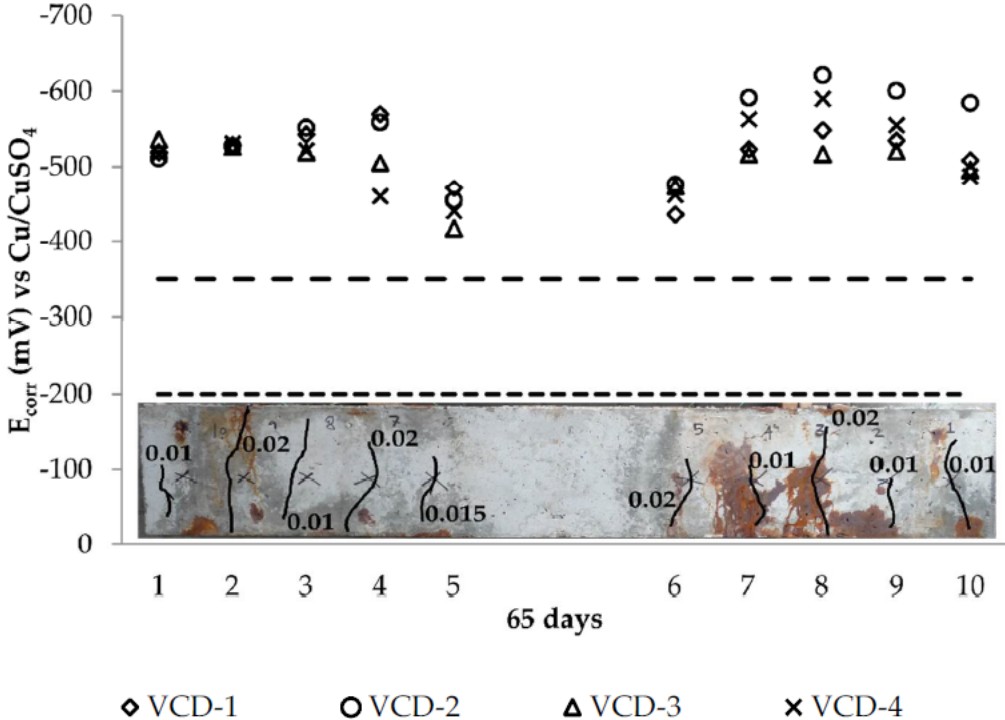

**Figure 13.** Electrochemical values at the locations of the shear reinforcements.

*4.3. Reliability of the Calculation and the Method to Register Mass Losses in the Cross-Section*

As previously mentioned, the reduction of the mechanical strength in the stirrup is proportional to its mass loss. As deterioration from corrosion progresses in the stirrups, the damage is irreversible. Therefore, it is important to clarify that, in practice, a 10% loss of cross-section from corrosion damage causes significant detachments and an extreme lack of safety that leads in most cases to demolitions rather than repairs. In accelerated tests, the attenuation factor plays an important role in the impact of the corrosion product, which is reflected in the behavior of the concrete under tensile stress. Because of this, the section cracks quickly and significantly [32] from the inability of the concrete pores to absorb the generated corrosion products quickly and efficiently. The opposite occurs in natural environments; i.e., as they are considered innocuous, the pores have a greater absorption capacity for corrosion products, and therefore significant losses of cross-section by shear can occur without cracks or significant detachments. Although the natural test is the best representation of reality in these cases, the accelerated test stimulates aggressive and destructive conditions. Therefore, any prediction based this it will always be conservative. This is important from the structural safety and preventive maintenance points of view. As discussed previously, in this study and based on the theoretical and experimental schemes, the indicators of severe deterioration were established. The indicators were used to evaluate and compare the experimental results through the relation between the experimental losses and the theoretical losses. The relation between the average experimental mass loss for each stirrup per beam and the loss calculated by Faraday's Law is shown in Figure 14. The upper line in the figure represents the theoretical mass loss estimated by the Faraday equation, with a value calculated at 65 days and equal to 50 g per stirrup. The bottom line represents the average experimental mass loss

per stirrup of 41.25 g corresponding to the values of the trend line with triangles in the same figure. These results indicate that the applied experimental methodology and the estimation of mass loss are related in an acceptable manner [33].

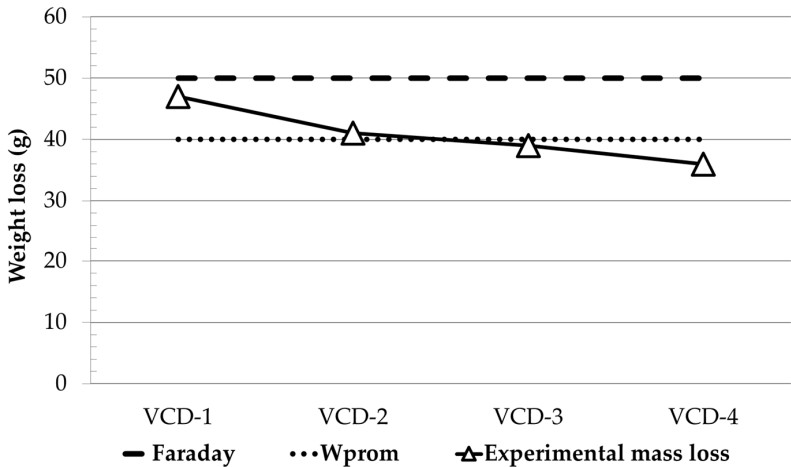

**Figure 14.** Ratio of experimental and calculated mass loss.

*4.4. Theoretical and Experimental Comparison of the Shear Damage (Vu/Vn)*

The method traditionally used in North America to design a structure is the ACI 318 code [27]. In this regard, the procedure in the structural behavior is designed in such a way that, in terms of durability, the beams will experience shear failure without bending. The shear strength of the concrete (Vc), shear strength of the stirrups (Vs) and nominal shear (Vn) were the results of the structural calculation of the beam, and the structural arrangement is presented in Figure 2. The damage induced in the stirrups through the proposed method to produce deterioration by corrosion generated mechanical repercussions, decreasing the ultimate shear strength from 20% to 30% for all VCD beams, according to the data in Table 5. Under these conditions, the data indicate that the partial reduction of the stirrups' section contributed more to the reduction of the ultimate shear strength of the VCD beams than of the VSD beams. Similarly, a change in the deflection at the midspan is exhibited in all the beams; therefore, the reduction in rigidity is caused by an increase in the fragility of the structure from internal pressures generated by the corrosion products. In addition, because all the beams contain the same variables (compressive strength, assembly, dimensions, deterioration), they can be compared. The data in Table 5 show a comparison between the values obtained experimentally and those estimated theoretically.

**Table 5.** Mechanical characteristics for VCD and VSD specimens.

| Beam | Diameter (mm) | | | Shear Strength (kN) | | | | |
|------|------|------|------|------|------|------|------|------|
| | Theor. | x̄ exp. | Critical | Vs | Vc | Vn | Vu | Vu/Vn |
| VSD-1 | 8.00 | 7.90 | 8.00 | 87 | 46 | 133 | 124 | 0.93 |
| VSD-2 | 8.00 | 7.90 | 8.00 | 87 | 46 | 133 | 126 | 0.95 |
| VCD-1 | 7.20 | 7.20 | 6.05 | 51 | 46 | 97 | 88 | 0.91 |
| VCD-2 | 7.20 | 7.20 | 5.90 | 49 | 46 | 95 | 84 | 0.88 |
| VCD-3 | 7.20 | 7.26 | 5.38 | 39 | 46 | 85 | 100 | 1.18 |
| VCD-4 | 7.20 | 7.30 | 5.40 | 41 | 46 | 87 | 98 | 1.12 |

## 5. Conclusions

The delimitation of the method and its verification will enable other studies to combine variables, such as the cross-section of concrete, different materials, and the conditions of exposure, in any type of environment, allowing comparison between results from different sources.

The method used in this work ensured that all the beams exhibited shear failure from diagonal stress when mechanically tested with almost 50% less deflection.

It was verified that, when using the proposed deterioration scheme, the values of $E_{corr}$ drastically decreased, generating severe corrosion in the shear reinforcements. The visual inspection of the beams confirmed the previous statements by the appearance of rust spots and cracks at the locations of the stirrups. The proposed method for the induction of deterioration by corrosion was reliable in terms of the reproducibility in corrosion potential values for each stirrup and with respect to the patterns and crack widths.

The previous general conclusions support the specific conclusions that follow, which can be readily compared with those of other studies in the literature.

The total mass loss of the stirrups was approximately 40–43 g per beam, which represents a decrease of approximately 11%.

The critical cross-sectional area, calculated with the experimental diameter with the greatest cross-sectional loss by corrosion of the deteriorated stirrup, proved to be a reliable value for predicting the ultimate shear strength of concrete beams deteriorated by severe corrosion. A reduction up to 30% in the shear strength of deteriorated beams relative to the beams without deterioration was found.

Finally, there is a correlation between the experimental and theoretical results based on the relation between the obtained results and the trends determined by the reference equations. This means that the procedure is reproducible and therefore adaptable to other type of conditions; thus, it is sustainable.

**Author Contributions:** Conceptualization, C.A.J.-A. and P.C.-B.; methodology, R.I.S.-I. and J.A.B.-M; formal analysis, P.C.-B. and C.A.J.-A.; investigation, R.I.S.-I. and J.A.B.-M.; resources, P.V.-T.; data curation, R.I.S.-I. and J.A.B.-M.; writing—original draft preparation, P.C.-B., C.A.J.-A., R.I.S.-I. and J.A.B.-M.; writing—review and editing, P.C.-B. and C.A.J.-A.; project administration, G.F.-S.M. All authors have read and agreed to the published version of the manuscript.

**Funding:** This research received no external funding.

**Acknowledgments:** The authors acknowledge the partial support of their Institutions and Conacyt. Two of the authors (R. I. Soto-Ibarra and J. A. Briceño-Mena) acknowledge the support of Conacyt through their Doctoral grants. Mercedes Balancán-Zapata helped both students with technical issues. The opinions and findings expressed on this manuscript are those of the authors and not necessarily those of the supporting Institutions. This paper is part of the collaboration and signed agreement between Cinvestav-Mérida and FIC-UANL.

**Conflicts of Interest:** The authors declare no conflict of interest.

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
