# Peer review of "Effect of Corrosion in the Transverse Reinforcements in Concrete Beams: Sustainable Method to Generate and Measure Deterioration"

_sustainability, doi:10.3390/su12198105_

Round 1

Reviewer 1 Report

The paper is interesting, but it seems to be written in a hurry: a whole series of details, that would make the work more usable for the reader, are missing. As an example:

  • it is never said what are, nor how are calculated, Vs Vc Vn Vu
  • only the crack pattern of the beam VCD-1 side A is shown
  • the load displacement curves are missing
  • mass loss per stirrups is indicated only for one beam (I think that different beams characterized by different ultimate load will be characterized by different mass losses)
  • the caption of Table 4 is "mix design", but evidently is incorrect
  • the % Chloride by weight of cement is indicated only for 2 beams

Moreover, the literature review should be improved. I recommend to read:

Andrade, C. (2019). Propagation of reinforcement corrosion: principles, testing and modelling. Materials and Structures52(1), 2.

Imperatore, S., & Rinaldi, Z. (2019). Cracking in Reinforced Concrete Structures Damaged by Artificial Corrosion: An Overview. The Open Construction & Building Technology Journal13(1).

Pedrosa, F., & Andrade, C. (2017). Corrosion induced cracking: Effect of different corrosion rates on crack width evolution. Construction and Building Materials133, 525-533.

Bossio, A., Lignola, G. P., Fabbrocino, F., Monetta, T., Prota, A., Bellucci, F., & Manfredi, G. (2017). Nondestructive assessment of corrosion of reinforcing bars through surface concrete cracks. Structural Concrete18(1), 104-117.

Reviewer 2 Report

In this paper a consistent method to generate and measure deterioration by corrosion in transverse reinforcements for concrete beams is presented and discussed. The added value of this research is its practical approach, large scale specimens and the correlation of experimental data with theoretical calculations.

The level of English is good, and the text is fluently legible.

The Abstract seems the weakest part of the paper. It does not seem to reflect the overall content of the paper and it is more like theoretical introductory part. It should be rewritten to more concise form, focusing on the research part of the paper.

The experiments are sufficiently described and all the relevant parameters are included. The results and discussion are clearly presented, with presentable Figures and concise Tables.

Conclusion creates a good summary of results, with the most important aspects and interpretations highlighted.

It is suggested to re-review relevant existing literature and (if applicable) refer also to findings from more recent publications – currently, the most recent referred publication is 5 years old.

Publication of this paper requires minor changes.

Round 2

Reviewer 1 Report

The reviewer thanks the authors for the effort made in improving the paper. On my point of view, now the article can be published,but only after a few more minor revision.
In detail:
1) the adopted current density should be better justified... some authors give a detailed indication on this aspect -see: a) Andrade, C. (2019). Propagation of reinforcement corrosion: principles, testing and modelling. Materials and Structures, 52(1), 2; b) El Maaddawy, T. A., & Soudki, K. A. (2003). Effectiveness of impressed current technique to simulate corrosion of steel reinforcement in concrete. Journal of materials in civil engineering, 15(1), 41-47.
2) the photograph in figure 3 should be enlarged and the caption of fig.3 should be more detailed (i.e.: on the left the detail of the assembled reinforcement, on the right the Section diagram with details of connections for corrosion induction
3) the red numbers in figure 6 are not readible, please enlarge it and use a different color
4) the texts in figure 8 are too small and little legible, should be modified their dimension
5) the horizontal axis in figure 10 should be limited to 15mm; why the beam VSD-1 has a different stiffness? this should be clarified in the text; moreover, it seems to me that the cracking load is different in the samples VCD2 and VCD3... is that right? this aspect should be highlighted and justified in section
6) the texts in figure 11 are too small and little legible, should be modified their dimension; moreover the representation of the beams should be the same of fig 12
7) the cracks and the red numbers in figure 13 are not readible, please enlarge it and use a different color
8) What does the triangular line in fig 14 represent?
